# A Pilot Study of A2NTX, a Novel Low-Molecular-Weight Neurotoxin Derived from Subtype A2 for Post-Stroke Lower Limb Spasticity: Comparison with OnabotulinumtoxinA

**DOI:** 10.3390/toxins14110739

**Published:** 2022-10-28

**Authors:** Ryuji Kaji, Ai Miyashiro, Nori Sato, Taiki Furumoto, Toshiaki Takeuchi, Ryosuke Miyamoto, Tomoko Kohda, Yuishin Izumi, Shunji Kozaki

**Affiliations:** 1Department of Clinical Neuroscience, Graduate School of Medicine, Tokushima University, Tokushima 770-8503, Japan; 2National Hospital Organization Utano Hospital, Kyoto 616-8255, Japan; 3Department of Rehabilitation Medicine, Tokushima University Hospital, Tokushima 770-8503, Japan; 4Laboratory of Epidemiology, Graduate School of Veterinary Sciences, Osaka Metropolitan University, Osaka 598-8531, Japan

**Keywords:** botulinum neurotoxin, subtype A2, A2NTX, onabotulinumtoxinA, clinical efficacy, safety, spasticity: modified Ashworth scale, Functional Independence Measure, hand grip, spread

## Abstract

All the currently used type A botulinum neurotoxins for clinical uses are of subtype A1. We compared the efficacy and safety for the first time head-to-head between a novel botulinum toxin A2NTX prepared from subtype A2 and onabotulinumtoxinA (BOTOX) derived from A1 for post-stroke spasticity. We assessed the modified Ashworth scale (MAS) of the ankle joint, the mobility scores of Functional Independence Measure (FIM), and the grip power of the unaffected hand before and after injecting 300 units of BOTOX or A2NTX into calf muscles. The procedure was done in a blinded manner for the patient, the injecting physician, and the examiner. Stroke patients with chronic spastic hemiparesis (15 for A2NTX and 16 for BOTOX) were enrolled, and 11 for A2NTX and 13 for BOTOX (MAS of ankle; > or = 2) were entered for the MAS study. Area-under-curves of changes in MAS (primary outcome) were greater for A2NTX by day 30 (*p* = 0.044), and were similar by day 60. FIM was significantly improved in the A2NTX group (*p* = 0.005), but not in the BOTOX group by day 60. The hand grip of the unaffected limb was significantly decreased in the BOTOX-injected group (*p* = 0.002), but was unaffected in the A2NTX-injected group by day 60, suggesting there was less spread of A2NTX to the upper limb than there was with BOTOX. Being a small-sized pilot investigation with an imbalance in the gender of the subjects, the present study suggested superior efficacy and safety of A2NTX, and warrants a larger scale clinical trial of A2NTX to confirm these preliminary results.

## 1. Introduction

Botulinum neurotoxins (BoNTs) are classified into seven immunologically distinct serotypes A–G [1,2]. Type A BoNT has been widely used clinically except for type B (rimabotulinumtoxinB or Myobloc^®^/Neurobloc^®^) [3,4]. Serotype A is now divided into subtypes A1–A8 with differing amino acid sequences [5,6] and only A1 toxins from Hall strain (e.g., onabotulinumtoxinA, OnaA or BOTOX^®^, abobotulinumtoxinA or Dysport^®^, incobotulinumtoxinA or Xeomin^®^) are available for clinical uses in the US [7,8]. Type A BoNTs exist in various molecular weights: LL (900 kD), L (500 kD), M (300 kD), and S (150 kD) toxins [3,9]. OnabotulinumtoxinA is a large molecular weight LL toxin, whereas incobotulinumtoxinA is a S toxin with low molecular weight [10,11]. It has been argued that incobotulinumtoxinA is less antigenic than onabotulinumtoxinA [11,12], whereas the latter might spread less than the former because of its large molecular weight [13,14]. 

We have developed a low molecular weight (150 kD) or S type BoNT of subtype A2 (A2NTX) produced by a unique strain of C. botulinum obtained from cases of infant botulism in Japan (Chiba-H) [15,16]. Subtype A2 toxins were found to enter the neuronal cells faster in vitro [17], and to be less diffusible to the contralateral limbs in vivo [18] than A1. A2NTX was also shown to be more efficacious per mouse LD50 unit in reducing muscle power in rats [18] and monkeys [19] than onabotulinumtoxinA (OnaA). The first-in-man clinical study [20] using injections into the extensor digitorum brevis (EDB) muscle indicated that A2NTX is around 1.54 times as potent as the same unit of onabotulinumtoxinA in reducing compound muscle action potentials (CMAPs), with similar duration of action to OnaA at the dose adjusted. 

Here, we present a small-sized pilot study of the head-to-head comparison of A2NTX and onabotulinumtoxinA (OnaA) on their efficacy and safety in treating post-stroke spasticity to explore its potential clinical utilities. This study is a part of a long-term clinical trial exploring the safety and tolerability of A2NTX [14], and all the entered subjects already had subtype A1 BoNTs (OnaA or S-toxin of A1 subtype, A1NTX) with a matched mixture and intervals from the previous injections. This study is registered (ClinicalTrials.gov identifier: NCT01910363, accessed 25 April 2022), and was published as abstracts [21,22] and as a proceeding reporting preliminary and partial data of the present study [23], but is the first to report the entire data package registered. 

The entire protocol was approved by the IRB of Tokushima University, Japan (date of approval: 31 March 2004, No. 2005–216, with revisions in 2006 and 2010). The study was conducted according to the guidelines of the Declaration of Helsinki. The procedure of administering to human subjects was before 2018, when the Japanese regulatory agency required all the test drugs to be manufactured in strict accordance with Good Manufacturer’s Practice (GMP). The last patient was injected prior to 2018 when the drug was considered legitimate. Informed consent was obtained from all patients in written form.

## 2. Results

### 2.1. Patients Breakdown

A total of 31 patients (16 for onabotulinumtoxinA and 15 for A2NTX) were enrolled after screening at day 0. All the subjects had previous injections of BoNTs [14], and the values of MAS had been small. We therefore set the criteria for MAS analysis as 24 patients who had greater values than 1 at least in the ankle flexors or greater than 0 for extensors (Full Analysis Set or FAS analysis). The rest of the outcomes were evaluated in the whole subjects (Intention-To-Treat or ITT analysis), although the number of subjects analyzed was variable in each measure.

The background of each group is depicted in Table 1, and there was no significant difference between the two except for the gender ratio and the cause of stroke (bleeding or infarct). The number of those who received OnaA and A1NTX in the past were matched (*n* = 15 or 16) for each group and intervals from the previous injections of OnaA and A1NTX were also similar (96.2 ± 2.5 days for OnaA, and 95.3 ± 1.8 days for A2NTX; mean ± SD). There was no subject who developed secondary unresponsiveness to BoNTs before entry.

The flow chart (Figure 1) shows the breakdown of excluded patients who did not meet the MAS criteria at day 0, and the total number of those who participated in MAS analysis (FAS) was 24 (Ona 13 and A2NTX 11). Since the range of modified Ashworth scale (MAS) would have been limited, if we included all patients, we set preconceived entry criteria of MAS being equal or more than two for analysis (ClinicalTrials.gov NCT01910363) as in other studies [24,25,26]. Excluded subjects whose MAS was zero in all extensor muscles. For the rest of analysis, all patients were included wherever their data were available (ITT).

Number of subjects may vary depending on availability of measure in ITT analysis.

Table 2 shows the summary of cases used for the main analysis of MAS (Modified Ashworth Scale) after exclusion of those who did not meet the criteria (Full Analysis Set or FAS). All the backgrounds here were balanced between the two groups.

### 2.2. Primary Outcomes

Table 3 shows a summary of MAS data (Full Analysis Set or FAS). Day 90 data were incomplete because of their reduced number of samples (*n* = 12 for OnaA and 10 for A2NTX), and are shown as being Appendix A.

As for changes of MAS (Figure 2), the A2NTX group showed significantly greater changes than the OnaA group at day 30 with modest levels.

Figure 3 summarizes the primary outcome of MAS/AUCs in both groups. Area-under-curves (AUCs) of MAS changes also demonstrated significantly greater values in the A2NTX-group at day 30, because of the same statistical nature as MAS changes at day 30. The co-primary of MAS/AUCs at day 60 showed greater values on the average, but there was no statistical difference between the two (*p* = 0.168). Day 90 data are incomplete and only Appendix A.

Each value indicates the sum of area-under-curves (AUCs) from day 0–30 (~day 30), day 0–60 (~day 60), and day 0–90 (Appendix A; ~day90). Vertical bars indicate standard errors.

### 2.3. Secondary Outcomes

Table 4 summarizes the results of the other secondary outcomes on the efficacy (mobility scores of Functional Independence Measure or FIM, and 10 m walking time) and safety (Manual Muscle Testing of the un-injected tibialis anterior muscle and the hand grip power on the unaffected side of the stroke as indicators of unwanted spread of action to a neighboring (tibialis anterior) and distant (hand grip) muscles).

As depicted in Figure 4, the A2NTX group, but not the OnaA group, showed highly significant functional improvements at day 60 in the within-group comparison before and after the treatment. Walking time for 10 m on the other hand showed no significant changes in both groups (Table 3).

As for the safety assessment, MMT of the tibialis anterior muscle showed no significant difference between the two groups, although the mean values were more reduced in the OnaA group than A2NTX. The hand grip power measurements showed a highly significant drop in the OnaA group at day 60, but not in the A2NTX-injected group (Figure 5).

Blood sampling (liver, renal functions, electrolytes, complete blood cells) showed no significant changes before and after the injections of BoNTs. Other clinical adverse effects were unremarkable as reported previously [14].

## 3. Discussion

Our previous clinical study on tolerability and safety of A2NTX in various muscle hyperactivities including dystonia and spasticity showed that doses up to 500 mouse LD50 units were well tolerated for longer terms, and the majority of the patients preferred A2NTX rather than onabotulinumtoxinA (OnaA) at the end of the test periods [14]. Despite the small sample size, the present study for the first time demonstrated that the efficacy of A2NTX as measured by MAS changes or their AUCs was higher than that of onabotulinumtoxinA at day 30 at amodest significance level. They were similar between the two groups at day 60.

Those at day 60 and day 90 were variable from case to case, possibly because of the differing efficacies of the patients’ individual rehabilitation, despite the constancy of their own methods throughout the study, and because of the limited number of subjects. Another reason for lack of statistical significance is the lack of a treatment effect at these timepoints. Since the patient backgrounds (Table 2; FAS) were comparable, at least the onset of clinical efficacy in MAS reduction seems to be faster for A2NTX than for OnaA.

Functional improvement as measured by the mobility scores of FIM was attained with a high significance level at day 60 in the A2NTX group, but not in the OnaA group. In ITT analysis (Table 1), there was male predominance and more bleeding as the cause in the OnaA group, and we must use caution in the interpretation of the results. It seems, however, unlikely that disease severity is different between the groups with the gender of subjects or the cause of stroke exerting notable influences, since the MAS scores and FIM levels were matched between the two groups. In addition, there was three patients with MAS being zero in the planter *extensor* muscle in the A2NTX group, who were excluded for FAS, but included in ITT analysis. It might be argued that these three cases contributed better outcomes in FIM. There was, however, no significant differences of pre-injection MAS of the extensor or flexor muscles between groups (Table 1), and it is unlikely that these specific cases contributed to any significant improvement.

As for safety, the hand grip of the unaffected limb decreased significantly at day 60 in the OnaA group, suggesting spread of the toxin effect to distant muscles, whereas that in A2NTX was unchanged as a whole. The increase in muscle power in some subjects who had A2NTX (Figure 5) may be because of already reduced grip strength due to the previous exposures to A1 toxins (OnaA and A1NTX), which could be reversed, since all patients had been treated with A1 toxins before. Again, the imbalance in gender may preclude the interpretation, but a larger number female subjects in the A2NTX group would have made A2 more susceptible to adverse effects such as distant spreading, because of the smaller body weights of the female. Nor is evidence that bleeding as the cause would enhance spreading of the effect to a distant muscle. Despite these limitations of the background imbalance, the present findings recapitulated the results of the previous clinical and in vivo studies of A2NTX, in that one mouse LD50 unit of A2NTX is as effective as 1.54 unit of onabotulinumtoxinA in reducing CMAPs [20], and that A2NTX spreads less of its action to distant muscles [14,27].

The major limitation of this study is the small number of subjects, which makes it an exploratory pilot study. The imbalance in gender and cause of stroke was also a limitation for ITT analysis. Although a modestly significant (*p* = 0.044) difference in MAS changes at day 30 may suffer from multiple comparisons, the present results may indicate the superior efficacy at day 30, since we set MAS changes up to 30 days or 60 days in AUCs as co-primary outcomes from the start (ClinicalTrials.gov NCT01910363). Despite being the secondary outcomes, FIM and hand-grip changes also endorsed higher safety and efficacy of A2NTX over OnaA with much higher significance levels. FIM was significantly improved only 60 days after injection in the A2NTX-injected group. This effect is probably the result of the combination of the BoNT injection and continued rehabilitation, which would require rather long periods for functional improvements.

The safety of BoNTs mainly depends on weakness in un-injected muscles [28], including respiratory ones [29,30]. As mentioned, BoNTs spread to other muscles locally in the adjacent muscles [20] or as reported by one group, and distantly through the nervous system [27,31]. Some of the local spread can be controlled by reducing the volume of injected BoNTs [32,33]. It is, however, unavoidable to see weakness in nearby muscles when large doses are injected [34]. It was demonstrated that OnaA can spread to the contralateral distant muscles trans-synaptically [35,36], although contradicting results were also reported [37]. These spreads through the spinal cord were more diffuse in OnaA than in A2NTX [27,36]. Much higher doses of BoNT can spread via the hematogenous route even with A2NTX [27]. The present study is the first to show a significant decrease of hand grip power in the contralateral upper limb after injection of OnaA into the lower limb, but with no drop after A2NTX. On the other hand, there was no significant difference in MMT, which reflects both local and trans-synaptic spread (Table 4), despite the larger drop of mean MMT for the OnaA group than A2NTX. This is possibly because the sensitivity of hand grip is higher than MMT.

Since all regulatory agencies stipulate that units are not interchangeable between different BoNT preparations, a comparison using the same mouse LD50 unit is not always appropriate for comparing safety and efficacy. In fact, one unit of A2NTX was shown to be 1.54 times as effective as one unit of OnaA [20]. While there seems to be better safety and tolerability profile of A2NTX than OnaA in the previous study [14], the exact assessment of efficacy and safety in man must be based on the ratio of efficacy and safety, such as the therapeutic margin used in the primate study of A2NTX [19].

It is generally argued that BoNT with high molecular weight is less diffusible [13,14]. Indeed, S toxin from A1 subtype BoNT or A1NTX was shown to be more diffusible than OnaA [14,38]. The reason for the even less spreading of A2NTX compared to OnaA is not clear, but it is conceivable that the higher affinity of A2 subtype BoNT to its receptors [17,39] outweighs its small molecular weight.

## 4. Future Perspectives

As for the clinical implication of the present study, the dose of the same 300 mouse LD50 units of OnaA and A2NTX suggested that A2 may be more efficacious and spread less than A1, despite its small sample size. Combined with the previous study where A2NTX was tolerable up to 500 u [14], the dose equivalent to 800 u of OnaA could be used successfully and safely used for treating spasticity if a larger clinical trial with botulinum toxin-naive patients fulfills the promise.

## 5. Conclusions

Despite the small-size and the difference in the gender ratio between the groups, the present pilot study suggested that A2NTX has an earlier onset of clinical efficacy as measured by MAS changes and higher efficacy in FIM and safety in distant spread than onabotulinumtoxinA, warranting a larger full-scale study.

## 6. Subjects and Methods

### 6.1. BoNT Preparations

OnabotulinumtoxinA (BOTOX^®^), marketed by GSK, Japan, Inc., was used. A2NTX (molecular weight: 150 k Dal) was produced and purified from the Chiba-H strain of Clostridium Botulinum, isolated from honey associated with cases of infant botulism [14,16] as previously described [18,40]. C. botulinum type A strains Chiba-H was cultured in a PYG medium containing 2% peptone, 0.5% yeast extract, 0.5% glucose, and 0.025% sodium thioglycolate by allowing them to stand at 30 °C for 3 days. M toxin was purified from the culture fluid by acid precipitation, protamine treatment, ion-exchange chromatography, and gel filtration. Each subtype of M toxin was adsorbed onto a DEAE Sepharose column equilibrated with 10 mM phosphate buffer, and eluted with a 0–0.3 M NaCl gradient buffer for the separation of A2NTX from the non-toxic component. Human serum albumin (Japan Red Cross, Japan) was added as an excipient. The toxicity of purified neurotoxin A2NTX, titrated by serial 2-fold dilution intraperitoneal injection measured as a mean 50% lethal dose (LD50), was 5.2 × 10^6^ LD50/mg protein [18]. A2NTX was stored in a deep freezer (<−70 °C) and thawed immediately before use.

### 6.2. Entry Criteria

Male or female patients suffering from lower limb spasticity after stroke with durations of more than 6 months, aged 40–79 years, and with no contractures were enrolled. Randomization was made as to age. Those who had regular rehabilitation therapy can be entered, but were not allowed to change their schedule in the period of assessment (day 0–60). For spasticity assessment, the Modified Ashworth scale (MAS; see below) of the ankle joint flexor or extensor must be more than or equal to 2 in the ankle flexor and more than 0 in the ankle extensor. These rather low MAS scores for entry were due to the fact that all patients had been treated with onabotulinumtoxinA (OnaA) or S-toxin of subtype A1 (A1NTX) previously [14]. Patients with botulinum toxin injections within 3 months of the study were excluded. Patients with serious hepatorenal dysfunction, cardiopulmonary failure, and those who cannot understand the instructions were also excluded.

### 6.3. BoNT Injection

Injections were performed by a single neurologist (RK), blinded as to the BoNT, with an EMG device (Clavis^®^, Medtronics Inc.), using an Ambu^®^ (744 75/10) monopolar lumen electrode of 75 mm length. BoNTs were first injected 150 LD50 units (diluted with 6 mL of saline) into the tibilais posterior muscle (TP); second, the needle was pulled out close to the skin, and the subject was injected in the medial gastrocnemius muscle (mGC) with 150 u (diluted with 6 mL of saline) with the same needle tract. The location was confirmed by EMG activation by passive stretch and electric stimulation causing movement of the foot (Figure 6).

The location of the needle insertion was determined as 5 finger-breadths distal to the tibial tubercle (TT) down the anterior margin of the tibia, and 2 finger-breadths posterior to the posterior edge of the tibia on the skin surface. The tip was directed toward the posterior aspect of the tibia. Caution was used to avoid vessels and nerves by checking the blood backflow from the needle and asking the subject to report any radiating pain along the foot. First, the tibialis posterior muscle was targeted, and then medial gastrocnemius was located just underneath the skin and confirmed by electric stimulation using the single needle tract.

### 6.4. Outcome Measures

#### 6.4.1. Modified Ashworth Scale (MAS)

The MAS was assessed as previously published [26,41]. Scales ranging from 0, 0.5, 1, 1.5, 2, 3, and 4 were analyzed as continuous values. Those of the ankle joint were measured in the ankle flexors and extensors, separately, and their changes after BoNT injections (day 0) were summed to represent the changes of the ankle joint MAS at day 30 and day 60. Day 90 was optional. Area-under-curves (AUCs) of MAS changes up to day 30 and up to day 60 were the primary outcome measures.

#### 6.4.2. Functional Independence Measure (FIM)

FIM was assessed as a measure of functional improvement [42,43]. Functional status of the patient’s mobility was assessed through physical examination or an interview from the care-giver by a blinded physiotherapist (TF) at day 30 and 60. The mobility part of FIM was calculated (minimum of 5, maximum of 35).

#### 6.4.3. 10 m Walking Time

Time (in seconds) required for the subject to walk for 10 m with or without assistance was established [44]. If assisted, the same assistance will be maintained throughout the study [24,44]. If the subject is unable to walk, the data were excluded for the analysis. The time was measured twice in a row, and the average of the 2 was processed.

#### 6.4.4. Manual Muscle Testing and Grip Strength

The MRC (Medical Research Council) scales (0, 1, 2, 3, 4−, 4+, 5) of manual muscle testing [43,45] of the antagonist (tibialis anterior) muscle were obtained at each visit by a physiatrist blinded as to BoNT (NS). The spread of BoNT action to the antagonist was thus assessed as a safety measure. Another safety assessment was the hand grip power [31] in kg torque of the side unaffected by the stroke. The hand grip was measured 3 times, and the average of the top 2 were calculated as the strength, providing evidence of the spread to distant muscles.

#### 6.4.5. Blood Sampling

At each visit (day −30, day 0, day 30, and day 60), blood samples were collected to evaluate complete blood cell counts, liver and renal function, and electrolytes.

### 6.5. Schedule of Visits

All the candidates for the study had a screening visit 30 days (day−30) prior to the 300 units BoNT injection for eligibility. When the subject was judged as eligible, blood sampling and a test injection of 50 units of BoNT into the flexor carpi radialis muscle on the affected side were performed (Figure 7). The purpose of this injection at day -30 was to test the new A2 toxin for any idiosyncratic adverse effects as required by IRB at a small dose in a different limb. On the injection day (day 0), MAS of the ankle flexors and extensors, the hand grip power of the unaffected upper limb, 10 m gait speed, and FIM were evaluated. The visit at day 90 was optional.

### 6.6. Statistical Analysis

Patients’ background differences were analyzed with the unpaired-t test or Fisher’s exact test. MAS was evaluated in the subjects who fulfilled the criteria of 0> and = or more than 2 at day 0). The area-under-curves (AUCs) of MAS changes of the ankle joint (the sum of the ankle flexor and extensor MAS changes) were analyzed. AUCs of day 30 (area over day 0–30) and of day 60 (areas over day 0–30–60) were calculated as primary outcomes. AUCs were compared with the Student-t test with a significance level of 0.05 (one-sided). Other parameters were analyzed, including all the subjects tested (ITT analysis): hand grip power in kg, gait time in seconds, and FIM, which were then compared within subjects using the paired-t test with a significance level of 0.05. Changes of MRC scales (ΔMMT) were compared with the Student-*t* test between A2NTX and OnaA. Since this study is a pilot study with preconceived primary outcomes, no corrections were made for multiple comparisons.

## Figures and Tables

**Figure 1 toxins-14-00739-f001:**
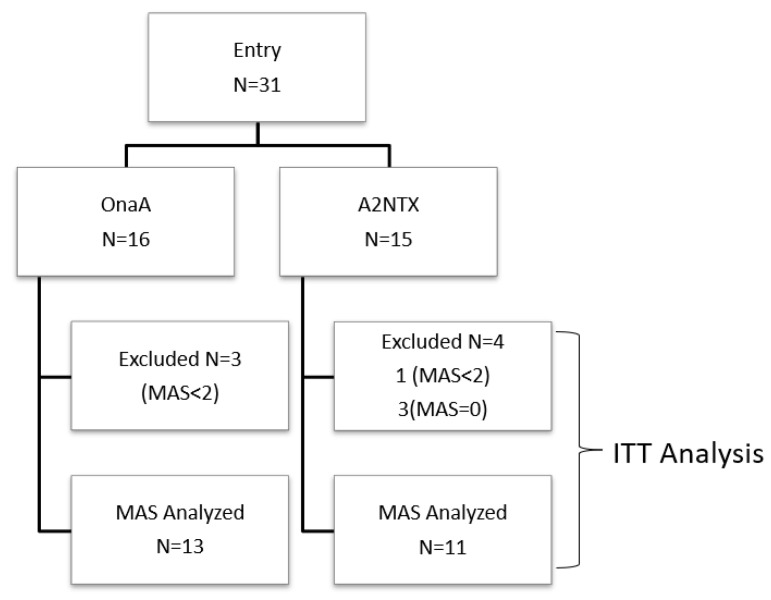
Flow chart for patients’ disposition.

**Figure 2 toxins-14-00739-f002:**
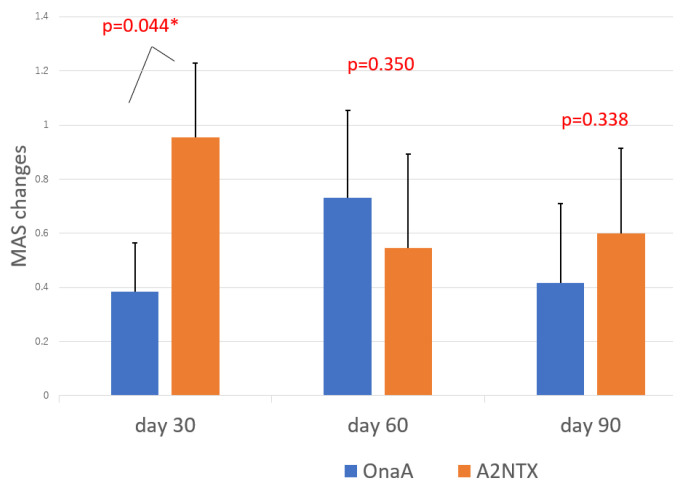
Changes of MAS. Each bar indicates the difference of MAS between day 0 and day 30, 60, and 90. Solid line on top indicates standard errors. * indicates significant difference.

**Figure 3 toxins-14-00739-f003:**
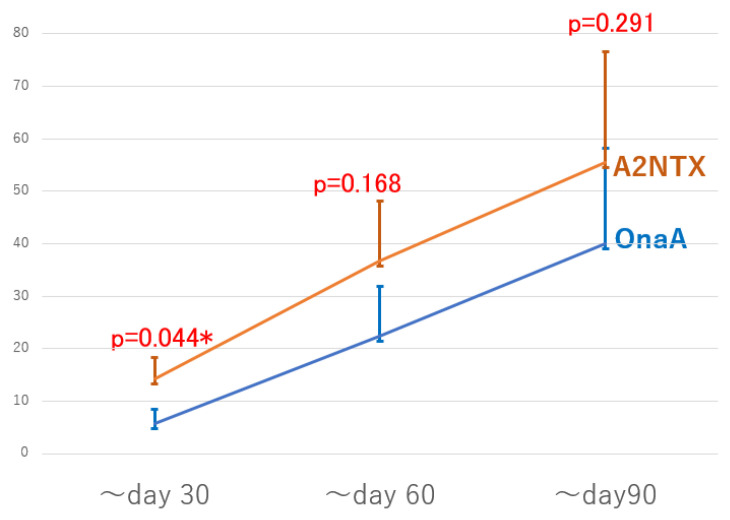
Changes of MAS/AUC (~day 30/~day 60: primary outcomes). * indicates significant difference.

**Figure 4 toxins-14-00739-f004:**
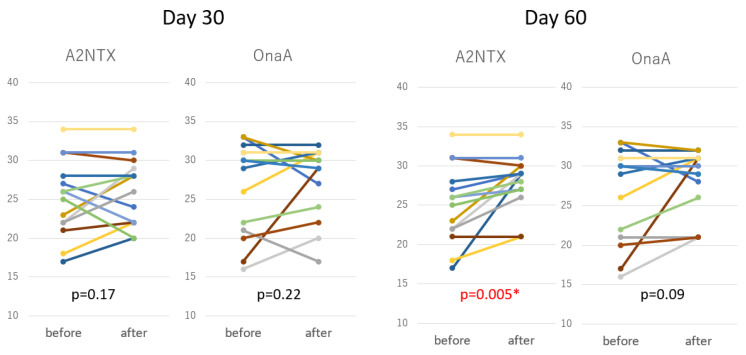
Functional Independence Measures (FIM) at day 30 and day 60. Each bar indicates individual subjects in different colors. A2NTX group showed highly significant improvement within group comparison at day 60 (*paired-t*). * indicates significant difference.

**Figure 5 toxins-14-00739-f005:**
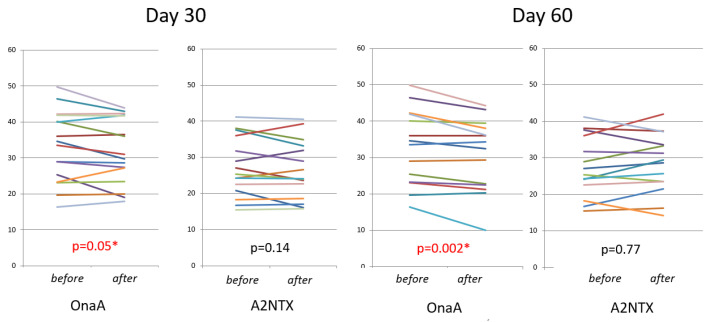
Hand Grip Power (in kg torque) of the unaffected side. Each bar indicates individual subjects in different colors. OnaA group showed significant drop of grip power at day 30 and 60 within group (*paired-t*). * indicates significant difference.

**Figure 6 toxins-14-00739-f006:**
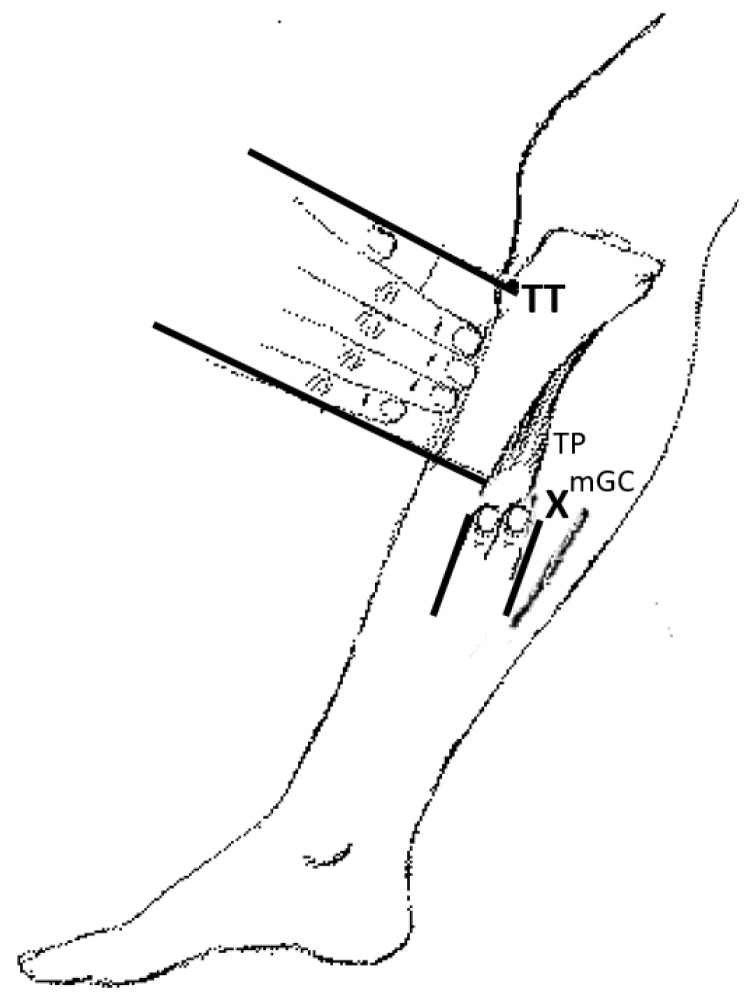
Method for injection.

**Figure 7 toxins-14-00739-f007:**
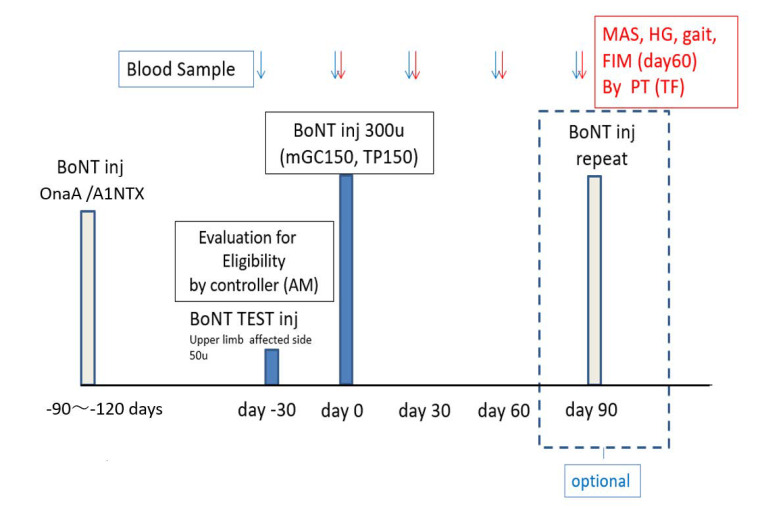
Summary of injection and evaluation schedule.

**Table 1 toxins-14-00739-t001:** Patients’ background (mean ± SD, range; ITT analysis ^1^).

	OnaA *n* = 16	A2NTX *n* = 15	Test	*p*
Sex (male/female)	15/1	10/5	Fisher’s exact	0.00014 *
Age (years)	65.8 ± 8.3 (52~78)	65.5 ± 9.2 (42~78)	*t*	0.88
Duration of illness (months)	98.1 ± 59.1 (22~240)	88.9 ± 66.0 (32~241.5)	*t*	0.58
Side of Paresis R/L	5/11	8/7	Fisher’s exact	0.23
Cause Bleeding/Infarct	12/4	9/6	Fisher’s exact	0.02 *
MAS (Ankle Flexion)	2.53 ± 0.85 (1~3)	2.53 ± 0.92 (1~4)	*t*	0.99
MAS (Ankle Extension)	0.94 ± 0.44 (0~2)	1.00 ± 0.93 (0~3)	*t*	0.81
Hand Grip Power (kg)	33.1 ± 8.2 (16.4~49.8)	27.2 ± 8.2 (15.5~38.1)	*t*	0.09
10 m Gait Time (s)	29.2 ± 24.5 (12.0~119.0)	32.5 ± 28.8 (9.3~114.2)	*t*	0.58
FIM (Full: 35)	26.9 ± 5.7 (17~33)	25.2 ± 4.8 (17~34)	*t*	0.41
MMT (Tibialis Anterior)	2.59 ± 1.38 (0–4)	2.40 ± 1.40 (0–4)	*t*	0.35

^1^ ITT: Intention-To-Treat, including all the data before exclusion for MAS criteria. * indicates significant difference.

**Table 2 toxins-14-00739-t002:** Patients’ background (mean ± SD, range; FAS analysis ^1^).

	OnaA *n* = 13	A2NTX *n* = 11	Test	*p*
Sex (male/female)	12/1	8/3	Fisher’s exact	0.08
Age (years)	64.1 ± 8.0 (52~78)	63.2 ± 10.3 (42~78)	*t*	0.88
Duration of illness (months)	88.7 ± 47.8 (22~192)	90.1 ± 53.5 (32~179)	*t*	0.94
Side of Paresis R/L	5/8	8/3	Fisher’s exact	0.54
Cause Bleeding/Infarct	10/3	8/3	Fisher’s exact	0.12
MAS (Ankle Flexion)	2.85 ± 0.55 (1~3)	2.73 ± 0.90 (1~4)	*t*	0.99
MAS (Ankle Extension)	1.08 ± 0.28 (1~2)	1.36 ± 0.81 (1~3)	*t*	0.81
Hand Grip Power (kg)	33.5 ± 10.4 (16.4~49.8)	28.0 ±8.1 (15.5~41.2)	*t*	0.09
10 m Gait Time (sec)	27.9 ± 27.3 (12.0~112.0)	37.5 ± 32.4 (9.3~114.2)	*t*	0.58
FIM (Full: 35)	26.7 ± 6.0 (16~33)	26.0 ± 4.8 (17~34)	*t*	0.41
MMT (Tibialis Anterior)	2.53 ± 1.39 (0–4)	2.45 ± 1.21 (1–4)	*t*	0.44

^1^ FAS: Full-Analysis-Set, after exclusion for MAS criteria.

**Table 3 toxins-14-00739-t003:** MAS data.

		A2NTX	OnaA	*p* Value
MAS changes	Day 30	0.95 ± 0.27 (*n* = 11)	0.38 ± 0.18 (*n* = 13)	0.044 *
	Day 60	0.55 ± 0.27 (*n* = 11)	0.73 ± 0.32 (*n* = 13)	0.350
	Day 90	0.60 ± 0.31(*n* = 10)	0.42 ± 0.29 (*n* = 12)	0.338
MAS changes/AUC *	**~Day 30**	**14.32 ± 4.10 (*n* = 11)**	**5.77 ± 2.71 (*n* = 13)**	**0.044 ***
	**~Day 60**	**36.82 ± 11.37 (*n* = 11)**	**22.5 ± 9.30 (*n* = 13)**	**0.168**
	~Day 90	55.50 ± 21.07 (*n* = 10)	40.0 ± 18.24 (*n* = 12)	0.291

* Bold figures indicate primary outcomes. Day 90 data are Appendix A.

**Table 4 toxins-14-00739-t004:** Secondary Outcomes (within-group comparison; paired-*t* test).

*Efficacy*	A2NTX	*p*	OnaA	*p*
FIM (Day 0)	25.1 ± 1.3 (*n* = 14)		26.7 ± 1.5 (*n* = 15)	
FIM (Day 30)	26.0 ± 1.2 (*n* = 14)	0.17	27.6 ± 1.2(*n* = 15)	0.22
FIM (Day 60)	27.9 ± 0.9 (*n* = 14)	**0.005 ***	28.3 ± 1.1 (*n* = 15)	0.09
10 m walking (Day 0)	31.6 ± 8.5 (*n* = 15)		26.4 ± 7.9 (*n* = 14)	
10 m walking (Day 30)	28.2 ± 7.1 (*n* = 15)	0.09	25.7 ± 7.0 (*n* = 14)	0.26
10 m walking (Day 60)	27.7 ± 6.4 (*n* = 15)	0.06	24.9 ± 7.3 (*n* = 14)	0.07
*Safety*				
Hand Grip (Day 0)	27.2 ± 2.1 (*n* = 15)		33.2 ± 2.5 (*n* = 16)	
Hand Grip (Day 30)	26.5 ± 2.1 (*n* = 15)	0.14	31.9 ± 2.3 (*n* = 16)	**0.05 ***
Hand Grip (Day 60)	28.4 ± 2.2 (*n* = 14)	0.77	30.7 ± 2.7 (*n* = 15)	**0.002 ***
ΔMMT (TA: Day 30)	−0.23 ± 0.19 (*n* = 15)	−0.56 ± 0.28 (*n* = 16)		0.17 ^‡^
ΔMMT (TA: Day 60)	0.30 ± 0.26 (*n* = 15)	−0.31 ± 0.4 (*n* = 16)		0.11 ^‡^

***** indicates significant values, and ^‡^ are results of between-group comparison (Student-t).

## Data Availability

Data used for the present study is available as a Appendix A.

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
