# Peer review of "A Pilot Study of A2NTX, a Novel Low-Molecular-Weight Neurotoxin Derived from Subtype A2 for Post-Stroke Lower Limb Spasticity: Comparison with OnabotulinumtoxinA"

_toxins, 2022, doi:10.3390/toxins14110739_

Round 1

Reviewer 1 Report (Previous Reviewer 4)

This is a revised submission, however, the conclusions the author concluded are still fully supported by the data. Especially, in line 254, the authors stated that “…suggested that A2 was more efficacious and less spreading than A1, despite its small sample size”. However, this can not preclude that the results the authors observed from hand grip were due to the imbalance of the sex of the subjects in the two groups. The earlier onset of clinical efficacy the authors observed was also barely significant with p=0.044, therefore, the authors' conclusions that the efficacy and safety are higher than onabotulinumtoxinA (line 261 -263) are not fully supported, especially the imbalance of sex in the two groups. Authors should consider changing the conclusion as inclusive and need a larger sample size with more balanced patient backgrounds between groups.

Other minor points: 1) Table 4, n numbers are different between FIM, 10m walking, hand grip, and DMMT, and also not matched to Figure 1.  2) Authors mentioned the supplementary data on day 90, however, there is no supplementary file. The non-published materials are a table without the table title and seem that they are some statistical analysis.

Author Response

This is a revised submission, however, the conclusions the author concluded are still fully supported by the data. Especially, in line 254, the authors stated that “…suggested that A2 was more efficacious and less spreading than A1, despite its small sample size”. However, this can not preclude that the results the authors observed from hand grip were due to the imbalance of the sex of the subjects in the two groups. The earlier onset of clinical efficacy the authors observed was also barely significant with p=0.044, therefore, the authors' conclusions that the efficacy and safety are higher than onabotulinumtoxinA (line 261 -263) are not fully supported, especially the imbalance of sex in the two groups. Authors should consider changing the conclusion as inclusive and need a larger sample size with more balanced patient backgrounds between groups.

- We have extensively revised the abstract and conclusion reducing the tone of the claim and placing sex difference as limitation.(line 28-31, 278-281).

Other minor points:

  • Table 4, n numbers are different between FIM, 10m walking, hand grip, and DMMT, and also not matched to Figure 1. 

- Some measures were not attainable in each item, and paired-t comparison were made one-to-one irrespective of the change in number. In Fig.1, the number of maximum subjects was shown, and this was added at the text and the legends.(line 107-108, 112).  

  • Authors mentioned the supplementary data on day 90, however, there is no supplementary file. The non-published materials are a table without the table title and seem that they are some statistical analysis.

- Day 90 was not followed in all the subjects, and was regarded as ‘supplementary’. Attached sheet is the statistical output on the background difference of the subjects, using Fisher’s exact test, which the reviewers were concerned. These are not intended for publication.

Reviewer 2 Report (Previous Reviewer 3)

In the present pilot study, the authors compared the clinical efficacy and safety of a novel botulinum toxin A2NTX prepared from subtype A2 and onabotulinumtoxinA (BOTOX) derived from A1 for spasticity, doing so they assessed the modified Ashworth scale (MAS) of the ankle joint, the mobility scores of Functional Independence Measure (FIM), and the grip power of the unaffected hand before and after injecting 300 units of BOTOX or A2NTX into calf muscles.

Unfortunately, this manuscript needs some improvements and corrections before publishing may be possible.

 General points:

 Please correct all spaces between the references numbers and words in the whole your manuscript.

Special points:

 Introduction

Please describe in your  Introduction section very exactly and very clear, which results in this study are completely new if compare with your previously nearly identical studies and publications [16, 17, 18].

Lines 32-42: please add multiple references at the end of each these sentences.  

Lines 43-58: please describe exactly all these studies.

Main part of the manuscript:

Please add to your manuscript the Future perspectives section. 

Discussion

Please describe in your  Discussion section very exactly and very clear, which results in this study are completely new if compare with your previously nearly identical studies and publications [16, 17, 18]. Please discuss these new results very exactly.

Lines 247-251: please add multiple references at the end of each these sentences.  

Materials and Methods

Lines 269-274: please add multiple references at the end of each these sentences.  

Author Response

In the present pilot study, the authors compared the clinical efficacy and safety of a novel botulinum toxin A2NTX prepared from subtype A2 and onabotulinumtoxinA (BOTOX) derived from A1 for spasticity, doing so they assessed the modified Ashworth scale (MAS) of the ankle joint, the mobility scores of Functional Independence Measure (FIM), and the grip power of the unaffected hand before and after injecting 300 units of BOTOX or A2NTX into calf muscles.

Unfortunately, this manuscript needs some improvements and corrections before publishing may be possible.

General points:

 Please correct all spaces between the references numbers and words in the whole your manuscript.

- We corrected the spaces between the reference numbers and words.

Special points:

 Introduction

Please describe in your  Introduction section very exactly and very clear, which results in this study are completely new if compare with your previously nearly identical studies and publications [16, 17, 18].

- Ref 16 is on the long-term safety and tolerability study, and ref 17, 18 are abstracts of the present study. Ref 19 is a review reporting preliminary and partial results of the study. These points were clarified line 70 - 72 in Introduction.  

Lines 32-42: please add multiple references at the end of each these sentences.  

- We added multiple references.

Lines 43-58: please describe exactly all these studies.

- We described these studies in more details.(line 54-63)

Main part of the manuscript:

Please add to your manuscript the Future perspectives section. 

- We added Future perspective section (line 269-275).

Discussion

Please describe in your Discussion section very exactly and very clear, which results in this study are completely new if compare with your previously nearly identical studies and publications [16, 17, 18]. Please discuss these new results very exactly.

- We revised Discussion in the light of the difference with our previous study (Takeuchi et al. 2021), and other aspects. (line195 -202).

 Lines 247-251: please add multiple references at the end of each these sentences.  

- We added multiple references.

Materials and Methods

Lines 269-274: please add multiple references at the end of each these sentences.  

- We added multiple references, and expanded the method in detail. (line 287- 300 ).

Reviewer 3 Report (Previous Reviewer 2)

This paper compares efficacy and safety of two different BTX formulations in lower limb spasticity.

The weakest aspect of this study - as already stated by the authors -  is the small size of the two patient groups. Although there was a statistically significant improvement of MAS changes at day 30 this improvement was only minor. In particular, based on the small sample size no conclusion about safety are appropriate. 

Fig. 2: How do the authors explain the unusual course of MAS changes after AN2NTX with improvement at day 30 , deterioration at day 60 and improvement thereafter, while the curve for ONA BTX is more conclusive?

Fig. 5: How do the authors explain the improvement of muscle force after AN2NTX injections? This is not logical but may have influenced the results.

Author Response

This paper compares efficacy and safety of two different BTX formulations in lower limb spasticity.

The weakest aspect of this study - as already stated by the authors -  is the small size of the two patient groups. Although there was a statistically significant improvement of MAS changes at day 30 this improvement was only minor. In particular, based on the small sample size no conclusion about safety are appropriate. 

- We revised the whole manuscript as a pilot study, and the conclusion is now that this warrants further studies with a larger size.(e.g. Abstract and Conclusion)

Fig. 2: How do the authors explain the unusual course of MAS changes after AN2NTX with improvement at day 30 , deterioration at day 60 and improvement thereafter, while the curve for ONA BTX is more conclusive?

- The reviewer’s point is well taken, but there was no statistical difference at day 60 (p=0.35), and day 90 is only supplementary. Therefore this difference in the time course between OnaA and A2NTX needs to be confirmed and discussed at further studies with larger sample size.

Fig. 5: How do the authors explain the improvement of muscle force after AN2NTX injections? This is not logical but may have influenced the results.

- As stated in lines 283-5 of the original manuscript (5.2 entry criteria), all the patients had been treated with A1 toxins before, and the increase in muscle power in some subjects may be because already reduced grip strength due to the previous exposures to A1 toxins (OnaA and A1NTX) could be reversed. This point was added in Discussion (line 218-221) in the new manuscript. 

Reviewer 4 Report (Previous Reviewer 1)

Thank you for addressing most of the comments.  Additional comments remain:

1. Regarding legitimate: please confirm that the last patient was injected prior to 2018 when the drug was considered legitimate

2. Table 1 and 2: please include baseline MMT for the limb TA that was tested in table 4

3. In statistical analysis: although noted in the discussion/limitations, please note that there was no correction for multiple corrections.

4. Figure 1 legend: I believe you mean "disposition" and not "discretion".  Please clarify

5. FAS is defined in Table 2, and not in Fig 1.  Since Fig 1 comes before Table 2, FAS should be defined in Fig 1, or the body of the text.

6. For "2.2 Primary Outcomes" - please stipulate whether this is the FAS or ITT.  

7. Regarding the sentence: "Those at day 60 and day 90 were variable from case to case, possibly because of the differing efficacies of the patients’ individual rehabilitation, despite the constancy of their own methods throughout the study, and because of the limited number of subjects": the interpretation of the data belongs in the Discussion and not the Results.  Also, another reason for lack of statistical significance is the lack of a treatment effect at those timepoints.  This should be covered in the Discussion.

8. Line 230: change "had-grip" to "hand-grip".

9. Line 237: change "or distantly through the nervous system[12,23]" to "or as reported by one group, distantly through the nervous system[12,23]. ".  Also, references 12 and 23 assert nervous system spread through the spinal cord, which appears not to be the case as the MMT are not different between A2NTX and OnaA

10: Line 239: Regarding "It is however unavoidable to see weakness in distant muscle when large doses are injected [29]": The cited reference uses the word "spread" to refer to "spread to surrounding structures" and "unintentional spread to nearby structures" (see Davis, 2020, citation 29).  However, the author of this paper uses it in the context of "distant".  Based on the literature, "distant" spread is not "unavoidable."  Please update.

Notably, the authors used a relatively dilute solution (25U/ml)/large volume (6ml) for spasticity.  

11: All regulatory agencies stipulate that units are not interchangeable.  This should also be written in the paper.  It can be included in the limitations section.

12: Relevant to the above, in the section on "BoNT preparations", the authors state: "A2NTX was stored in a deep freezer (<-70°C) and thawed immediately before use."  However, the authors need to state how the A2NTX was formulated before injection into subjects.  In other words, the reader needs to understand the excipients added to the raw material.

13: Line 273, the potency of A2NTX is stated as "was 5.2 ×10LD50/mg protein".  I think that a number is left out as the potency is expressed in the manuscript as a ' (see highlight).

Author Response

Thank you for addressing most of the comments.  Additional comments remain:

- Thank you for your kind reviewing!

  1. Regarding legitimate: please confirm that the last patient was injected prior to 2018 when the drug was considered legitimate

- We added this at Introduction ( line 78-79 ).

  1. Table 1 and 2: please include baseline MMT for the limb TA that was tested in table 4

- We added baseline values of MMT in Table 1 and 2.

  1. In statistical analysis: although noted in the discussion/limitations, please note that there was no correction for multiple corrections.

- This was added at Method Section (line 391-392).

  1. Figure 1 legend: I believe you mean "disposition" and not "discretion".  Please clarify

- We changed the wording.

  1. FAS is defined in Table 2, and not in Fig 1.  Since Fig 1 comes before Table 2, FAS should be defined in Fig 1, or the body of the text.

- We included the definition of FAS and ITT at the first paragraph of Results section in the body of the text (line 85-90 ).

  1. For "2.2 Primary Outcomes" - please stipulate whether this is the FAS or ITT.  

- We included FAS at the first paragraph of 2.2 (line 127).

  1. Regarding the sentence: "Those at day 60 and day 90 were variable from case to case, possibly because of the differing efficacies of the patients’ individual rehabilitation, despite the constancy of their own methods throughout the study, and because of the limited number of subjects": the interpretation of the data belongs in the Discussion and not the Results. 

- We moved these into Discussion (line 203- 208).

Also, another reason for lack of statistical significance is the lack of a treatment effect at those timepoints.  This should be covered in the Discussion.

- We added this sentence to Discussion (line 205- 206 ).

  1. Line 230: change "had-grip" to "hand-grip".

- We revised.

  1. Line 237: change "or distantly through the nervous system[12,23]" to "or as reported by one group, distantly through the nervous system[12,23]. ". 

- We revised. (line 243)

Also, references 12 and 23 assert nervous system spread through the spinal cord, which appears not to be the case as the MMT are not different between A2NTX and OnaA

- While this is an important point, there seems to be larger drop of MMT (Table 4) for OnaA group than A2NTX, despite the lack of significance. This is possibly because the sensitivity of hand grip is higher than MMT. We added this point in Discussion. (line 252-255)

10: Line 239: Regarding "It is however unavoidable to see weakness in distant muscle when large doses are injected [29]": The cited reference uses the word "spread" to refer to "spread to surrounding structures" and "unintentional spread to nearby structures" (see Davis, 2020, citation 29).  However, the author of this paper uses it in the context of "distant".  Based on the literature, "distant" spread is not "unavoidable."  Please update.

- We deleted ‘distant’, and used ‘nearby’. (line 245)

Notably, the authors used a relatively dilute solution (25U/ml)/large volume (6ml) for spasticity.  

- While we agree on this point with the reviewer, it is not a very high dilution, considering the volume of the muscles.

11: All regulatory agencies stipulate that units are not interchangeable.  This should also be written in the paper.  It can be included in the limitations section.

- We added a paragraph of the limitation of using mouse LD50 units for comparison (line 256 - 262).

12: Relevant to the above, in the section on "BoNT preparations", the authors state: "A2NTX was stored in a deep freezer (<-70°C) and thawed immediately before use."  However, the authors need to state how the A2NTX was formulated before injection into subjects.  In other words, the reader needs to understand the excipients added to the raw material.

- We exapnded these in detail at Method section (line 287-300 ).

13: Line 273, the potency of A2NTX is stated as "was 5.2 ×10’ LD50/mg protein".  I think that a number is left out as the potency is expressed in the manuscript as a ' (see highlight).

- We revised this to 106, in the Method section.(line 299).

Round 2

Reviewer 1 Report (Previous Reviewer 4)

Authors claimed that some measures were not attainable in each item. For their primary outcome (MAS), they exclude 3 subjects in OnaA group and 4 subjects in A2NTX group. But this full analysis set did not apply to the analysis of secondary outcomes. Authors need keep the same criteria for both outcomes, or they should discuss why those excluded subjects in primary outcome analysis being included in the secondary outcomes. for example, there are three subjects in A2NTX group with MAS=0 (being excluded), and those may contribute to the better secondary outcomes and generate the bias. 

Author Response

Thank you for the review:

 Authors need keep the same criteria for both outcomes, or they should discuss why those excluded subjects in primary outcome analysis being included in the secondary outcomes. for example, there are three subjects in A2NTX group with MAS=0 (being excluded), and those may contribute to the better secondary outcomes and generate the bias. 

- We added in Results Section (line 97-98) that these MAS 0 subjects are all in ankle extensors, and not in injected flexor muscles. We also discussed this in line 206-211.

Reviewer 2 Report (Previous Reviewer 3)

Thank you for all corrections.

Author Response

We corrected the sentence in line 126-127 for redundancy of 'significance'. 

Reviewer 3 Report (Previous Reviewer 2)

none

Author Response

We corrected line 126-127 for redundancy of the word 'significance'. 

This manuscript is a resubmission of an earlier submission. The following is a list of the peer review reports and author responses from that submission.

Round 1

Reviewer 1 Report

The authors seek to distinguish 2 subtypes of BoNT/A with this clinical study.  Unfortunately, there are several confounds that make the results difficult to interpret.  Specific comments or questions are listed below, by Line.

54: what is the significance of the human subjects consent "fully legitimate until 2018".  What happened after 2018?

62: in addition to sex, the study is not balanced by side of paresis.  This is important for the assessment of hand grip power.  The lack of balance needs to be called out in the results and a statement of limitations for this manuscript.

69: We should see a table similar to Table 1 for the ITT analysis.  I am assuming that all of the subsequent tables list data for just the ITT population.  Please confirm.

Tables and figures: all tables and figures need to have the number of subjects analyzed, i.e., the "n".  Also, all p values need to be listed.  For instance, Figure 2 has only a p value for D30 and not D60 or D90.  Fig 3 is missing the value for D90, etc.

91: lack of consistency of rehabilitation use over time is a limitation of the analysis.  This should be addressed later in the Ms.  Also, some table needs to list the number of patients in each group that had rehabilitation treatment during the study  

116: Please address in the discussion that manual muscle testing was not standardized for dominant/non-dominant hand.  This is a limitation

152: please address why you believe that the MAS findings at D30 were not evident at subsequent visits.

174: Please note that there is recent data that has reappraised the trans-synaptic spread of toxin (see Cai, 2017).  Please cite this and note that the initial studies are not conclusive.

207: Prior treatment with botulinum toxin becomes a confound.  Notably, 3 months is probably not sufficient for the full effect to wear off.  This should be noted in the limitations of the study.  In addition, the tables should list the number of patients in each group that were previously treated, and this covariate should be taken into account in an exploratory analysis.  

211: what was the volume injected?

219: has the sum of the MAS's been validated as a valid outcome variable?  Nevertheless, the individual MAS scores should be presented in a table by muscle group for the analysis population, with the relevant statistics.  

230: in the tables of clinical characteristics, we need a summary of the number of patients that were ambulatory by treatment intervention

250: (a) what was the purpose of the eligibility injection on D-30? (b) a visit schedule of hand grip testing is needed.  It is ambiguous whether D-30 is considered D0 for hand grip testing. 

The paper needs a list of limitations, as the design is such that one can not draw firm conclusions from this set of data. 

Reviewer 2 Report

This paper compares to different botulinum toxins in the treatment of spasticity.

The paper has a number of points that need to be addressed: the sample size is much to small to allow any final conclusions and the patients enroled are very heterogenous. The groups differ in etiology (many more bleedings in OnaBot), there is no sufficent information about the clinical picture and details of spasticity. Where there any contractions? What was the dilution of toxins? The discussion about transsynaptic transport of BTX is very speculative. With regard to hematogenic spread were there any neurophysiological testings done (single fiber EMG, nerve conduction studies)? Any differences in swallowing?

In summary, based on the heterogenous group of patients and the small number of participants in the the two arms of the study, is does not seem justified to draw any major conclusions.

Reviewer 3 Report

The authors presented a pilot, double-blind, controlled study of head-to-head comparison of onabotulinumtoxinA and A2NTX on their efficacy and safety in treating post-stroke spasticity to explore its potential clinical utilities.

This manuscript is interesting; nevertheless needs substantial improvements and corrections before publishing may be possible.

General points:

Please do your List of References and citations in the whole manuscript according to “Toxins”.

Please correct your List of  References: your list end with the number “18”, but the citations  in your manuscript end with “43”.  

Keywords

Please add also to keywords: A2NTX.

Special points:

Introduction

This manuscript should be substantially improved, i. e., by substantial references in the field:

Lines 30-40: please add multiple references at the end of all these sentences.

Lines 43-45: please describe very exactly all these studies.  

Please describe very exactly all studies, which already done with A2NTX  in animal models and in patients. Also, please describe exactly the studies 41-43.

Figures

Figure 1: please add an exactly description to this Figure as a Legend.

Figure 2: please add an exactly description to this Figure as a Legend.

Figure 3: please add an exactly description to this Figure as a Legend.

Figure 4: please add an exactly description to this Figure as a Legend.

Figure 5: please add an exactly description to this Figure as a Legend.

Subjects and Methods

 Lines 197-199: please describe this very exactly.

Please add to your manuscript body a Figure demonstrates the injections points into the tibialis posterior muscle and medial gastrocnemius muscle.

Please add appropriate references to your 10 m walking time test.

Please add appropriate references to your Manual muscle testing and Grip strength.

Please add to your manuscript a Future perspectives section.

Please describe your Statistical Analysis section more exactly.

Reviewer 4 Report

The authors described a small pilot study on clinical efficacy and safety data of botulinum neurotoxin type A2 strain, and compared it with OnaA. There are a few things author need to clarify:

1.       From the schedule of visits in the subjects and methods section, the authors claimed that the BoNT injection elsewhere 90 days or more before the BoNT injection in this trial is optional. Did they track how many have the BoNT injection 90 days or more before the trial? How are those subjects with pre-exposure to BoNT distributed in the two groups (A2NTX and OnaA)? It is known that the repeated injections of BoNT may generate antibodies that reduce the efficacy of BoNT. So it is important to include this information, and if the distribution is uneven, then it may contribute to the reduced efficacy.

2.       Page 8, line 254, should it be Figure 6?

3.       The authors mentioned supplementary data, however, there are no supplementary data uploaded.

4.       The authors cited references up to 43, but the references section only includes 18 references.

5.       While the authors pointed out the significance of MAS changes at day 30 only minimum (p=0.044), in the conclusions section, the authors claimed the present results suggested A2NTX has superior clinical efficacy. Given the sample size is so small, and the p-value, this conclusion is misleading. The superior efficacy needs to be tested in a larger clinical trial.